# Application of Plasma Bridge for Grounding of Conductive Substrates Treated by Transferred Pulsed Atmospheric Arc

**Dariusz Korzec * , Markus Hoffmann † and Stefan Nettesheim**

Relyon Plasma GmbH, Osterhofener Straße 6, 93055 Regensburg, Germany
* Correspondence: d.korzec@relyon-plasma.com
† Current affiliation: OPTEL Group GmbH, Kapellenstr. 11, 85622 Feldkirchen, Germany.

**Abstract:** An atmospheric pressure plasma jet (APPJ) sustained by a pulsed atmospheric arc (PAA) transferred on an electrically conducting surface was operated with a mean power of 700 W, a pulse frequency of 60 kHz, and a gas mixture of $N_2$ and $H_2$ with up to 10% $H_2$, flowing at 30 to 70 SLM. It was shown that the plasma bridge ignited between the grounded injector and electrically conducting and floating substrates can be used for electrical grounding. This allowed for arc transfer on such substrates. The plasma bridge was stable for Argon flow through the injector from 3 to 10 SLM. Its length was between 5 and 15 mm. The plasma bridge current was 350 mA. The copper contact pads on an alumina electronic board were treated using the plasma bridge sustained by Ar injection for grounding. First, an oxide film of about 65 nm was grown by a compressed dry air (CDA) plasma jet. Then, this film was reduced at a speed of 4 $cm^2/s$ by forming gas 95/5 (95% of $N_2$ and 5% of $H_2$) plasma jet.

**Keywords:** atmospheric pressure plasma; atmospheric pressure plasma jet; arc; transferred arc; pulsed atmospheric arc; plasma bridge; oxide reduction





## 1. Introduction

The atmospheric pressure plasma jet (APPJ) in its numerous variations [1,2] is a kind of cold atmospheric plasma (CAP) or atmospheric pressure plasma (APP) [3–6] broadly used in research and industry. The low-temperature arc jet [7,8] has a high potential for material processing due to its high local plasma density. The popular way of arc stabilization in low-temperature arc jets is a gas vortex [9]. The physical [10], electrical [11], and material [12] properties of such jets have been investigated. Many applications have been described, e.g., the modification of the surface of polymers for the improvement of adhesion [13], for example polyethylene [14], glass-fiber-reinforced polypropylene [15], or polydimethylsiloxane (PDMS) [16]. Furthermore, metal surfaces can be treated [17]. Surface modification for the hydrophilic property of stainless steel treated by an atmospheric pressure $N_2$-$O_2$ plasma jet was demonstrated in [18]. Varnished or polymer-coated metal surfaces can also be successfully treated [19]. A material increasingly interesting for surface treatment is glass [10]. Other applications are oxidation and rapid annealing [20].

The plasma generator used in this study belongs to the pulsed atmospheric arc plasma jets (PAA-PJs). Its discriminating feature is the generation of the arc via HV pulses in the kHz range with voltages up to 15 kV for ignition and in the range of 500 V to 3000 V for sustaining the plasma. Recently, its physics was investigated by laser scattering techniques [21] and optical spectroscopy [22]. The influence of the pulse amplitude and frequency on PAA-PJ's properties was examined [23].

The PAA-PJ can be operated in diffuse or focused plasma mode. The broadest application of diffuse mode is the activation of different surfaces to increase the surface free energy (SFE) of polymers for the improvement of the paintability or gluing properties or for casting. The enhancement of the bonding properties of pressure-sensitive adhesives on coatings

of white goods by means of atmospheric pressure plasma treatment has been demonstrated [24]. Another example is the surface modification of carbon fibers [25]. A strong increase of wetting, expressed as a decrease of the contact angle, resulting in the widening of printed electric contacts was observed after nitrogen plasma treatment of gas, PI, and PET before aerosol jet printing [26]. The improvement of the mechanical shear strength of glued joints on PAA-PJ-treated aerosol-jet-printed pads has been documented [27]. The suitability of the PAA-PJ for bacteria inactivation on temperature-sensitive surfaces was demonstrated on an example of *geobacillus stearothermophilus* spores [28].

The high energy density in the arc zone allows for the use of the PAA-PJ for coating processes. A 1.29 nm/s deposition rate of zinc oxide was demonstrated using a nebulized $ZnCl_2$ solution sprayed into the downstream of the nitrogen plasma jet [29]. The PAA-PJ can be used for low-density polyethylene coating [30], fluxing of printed circuit boards [31], coating wood with polyester [32] or $TiO_2$ [33], and coating of bismuth oxide circular droplets [34].

The term focused plasma mode is used for the operation of the PAA-PJ with the HV arc transferred from the grounded nozzle to a grounded, electrically conducting substrate. Under such conditions, a dense plasma is produced directly at the substrate surface. The transferred arc has been proven to be efficient for the cleaning, delubrication, oxide reduction, surface roughening, or depainting of metal surfaces. The mechanism of adhesion improvement is mainly chemical, but also, an increase in the surface roughness is involved. An additional advantage of the transferred arc is that no erosion of the nozzle occurs, and consequently, the lifetime of the nozzle can be prolonged by an order of magnitude [35]. The applicability of this operation mode is limited by the need for the connection of the substrate to the electrical ground which is not always possible. This study showed how an electrically floating surface charged up by a transferred arc can be grounded using a plasma bridge. Originally, the term plasma bridge referred to the low-pressure plasma used for the neutralization of an ion beam [36–39]. However, in this study, it means the gaseous discharge ignited at atmospheric pressure in a gas with a low breakdown voltage such as argon for establishing a highly conductive electric connection between the substrate and the grounded gas injector. In this paper, the basic properties of such a plasma bridge are discussed. The reduction of oxidized copper contact pads distributed on a ceramic plate demonstrated the technological applicability of such a plasma bridge.

## 2. Materials and Methods

### 2.1. Materials

To sustain the PAA-PJ, CDA, forming gas 95/5 (FG 95/5), forming gas 90/10 (FG 90/10), and the nitrogen 5.0 of Linde AG were used. The FG 90/10 was mixed with nitrogen 5.0 to reach the nitrogen–hydrogen gas mixtures with a percentage of hydrogen from 1 to 10%. The FG 95/5 was used for process development, because 5% of hydrogen is the highest percentage at which the $N_2$-$H_2$ gas mixture is non-flammable. The plasma bridge was generated using either Varigon H2 (98% Ar, 2% $H_2$) or the argon 5.0 of Linde AG.

The electronic boards provided by AB Mikroelektronik GmbH were used as test substrates for oxidation and reduction process demonstration (see Section 3.4). They are $Al_2O_3$ plates with a size 175 mm × 120 mm × 1 mm and are covered with non-connected square contact pads made of copper. The squares have different areas. There are 88 contact pads with an area of 175 $mm^2$, 84 with 32 $mm^2$, and 112 with 8 $mm^2$.

### 2.2. Setup for Plasma Bridge Investigation

Figure 1 shows the setup implemented for the investigation of the plasma bridge. A stainless steel plate with a size of 100 mm × 25 mm × 2 mm was used as an electrically floating substrate. The distance between this plate and the plasma nozzle can be varied. The current flowing through the plasma bridge was measured on the wire grounding the injector tube. The Tektronix TCP202 DC coupled current probe connected to the oscilloscope Tektronix DPO3000 was used for this measurement. The mass flow controller (MFC) Model

8626 of Fa. Bürkert for a max. 100 SLM dosed the gas for the plasma jet and for the plasma bridge.

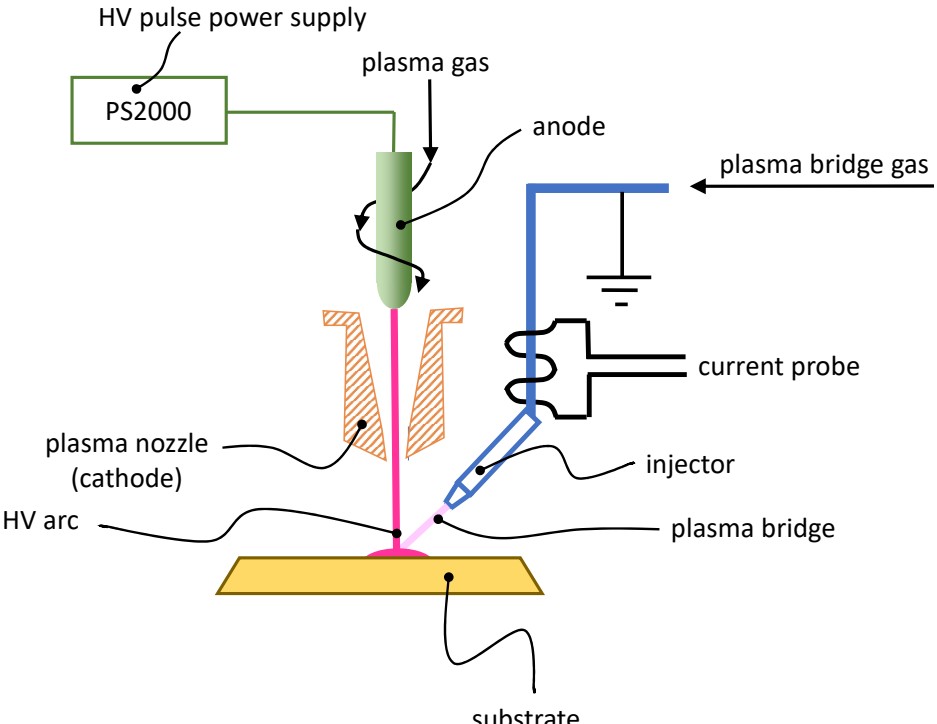

**Figure 1.** Setup for plasma bridge investigation.

For the generation of the PAA-PJ, the plasma generator PG31 [40] of relyon plasma GmbH was used. The cross-section of this device was shown and its operation principle was explained in [30]. The arc needed for plasma generation was sustained between the positively biased inner electrode (anode) and the outer edge of the grounded plasma nozzle (cathode) by HV pulses, supplied by a commercial power supply PS2000 OEM [41], connected via a 10 m coaxial HV cable to the plasma generator. The frequency of the pulses can be set between 40 and 65 kHz. The relative power level can be set in percent. The mean values of the voltage, current, and power were measured internally at the HV output and were available over the CANopen interface. After arc ignition with a voltage in the range of 15 kV, the power was controlled, resulting in the operation voltage between 500 and 3000 V, depending on the operating conditions (type of gas, gas pressure, gas temperature, frequency). This system is especially suitable for this investigation and process, thanks to the positive polarization of the inner electrode, allowing for the cathodic cleaning [42] of the grounded substrates.

The swirl-type gas flow stabilized the HV arc along the axis of the plasma nozzle. The PG31 is equipped with the nozzle A450 of relyon plasma GmbH. The nozzle is made of copper. The diameter of its gas outlet orifice is 4.2 mm.

The plasma bridge gas was injected into the plasma zone through an external injector (see Figures 1 and 2) made of brass, with an orifice diameter of 1.5 mm. The axial and radial position and the angular orientation of the injector are adjustable.

The temperature was statically measured using a K-type temperature sensor embedded in $Al_2O_3$ tubing with an outer diameter of 1.2 mm to avoid electromagnetic interference in the sensor electronics by arcs transferred onto its surface.

The Nikon D5200 digital camera allowed taking 5 pictures per second or making videos at 60 frames per second.

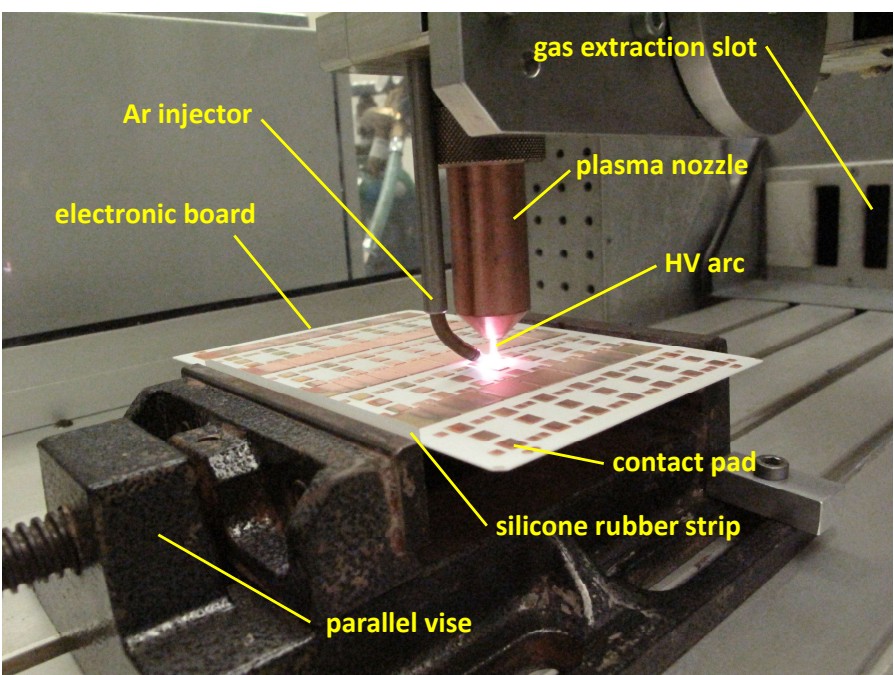

**Figure 2.** The copper contact pads distributed at the alumina plate are grounded by the Ar plasma bridge during processing with the PAA-PJ. The plasma generator is fed with FG 95/5.

### 2.3. Plasma Processing System

The architecture of the processing system used for the application example was essentially the same as for the plasma bridge characterization, as shown in Figure 1.

The main difference was the application of the xyz-robot ISEL GFV 44/33 for the motion of the plasma generator PG31, as shown in Figure 2. The robot was controlled by a personal computer. The position of the plasma head can be set in the Cartesian coordinates with a resolution of 1 mm. An optional manual adjustment of the tilt angle is possible, but not used for this study. The PG31 remained always perpendicular to the alumina plate. The substrates were fixed horizontally between the jaws of the parallel vise. To avoid thermal damage due to the thermal expansion of the ceramic board, one of the vise jaws was covered by a silicone rubber strip, as shown in Figure 2. The plasma head moved parallel to the vise jaws and perpendicular to the plane, in which the plasma bridge was established with a maximum speed of 300 mm/s.

The system was enclosed in a box with transparent walls and with gas extraction. On the back side of the box, the slots for gas extraction can be seen.

## 3. Results and Discussion

### 3.1. Diffuse Plasma Mode

#### 3.1.1. Structure of the PAA-PJ

Figure 3a shows the free-standing PAA-PJ sustained with FG 95/5 as the ionization gas. Two zones can be recognized: the shorter one, consisting of narrow filaments recognizable as HV arcs, and the longer, diffuse one, depicted as diffuse plasma in the figure.

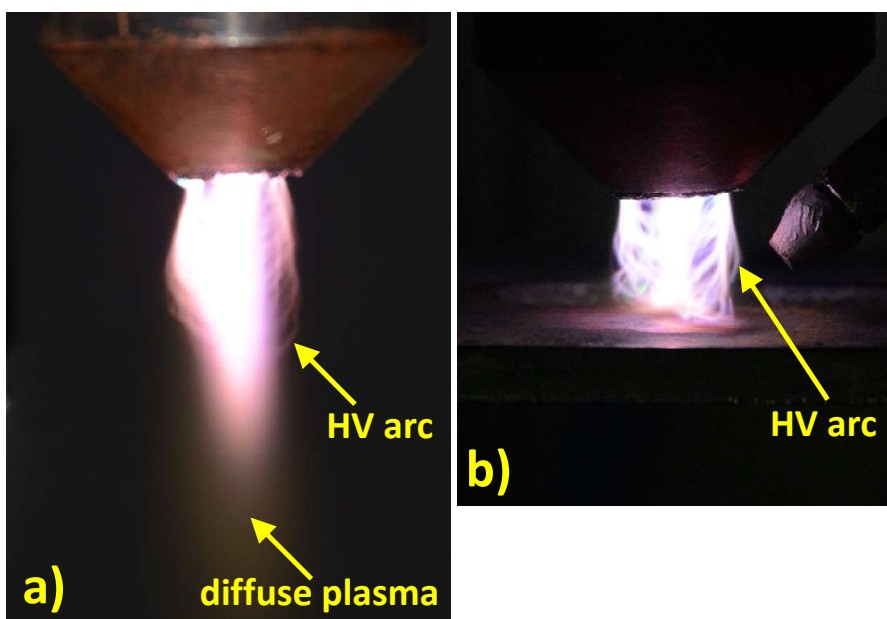

**Figure 3.** The plasma jet with no arc transfer (**a**) in free space and (**b**) directed on a non-grounded metallic substrate. Conditions: forming gas 95/5, flow 60 SLM, frequency 62 kHz, power 100%.

After plasma ignition, the HV arc is established between the tip of the internal anode and the nozzle lip, as shown in Figure 4a. Due to the swirling motion of the air, a drag on the cathodic arc foot exists, which causes its movement around the nozzle lip. The rotation speed reaches a couple of thousand rotations per second. During one rotation cycle, 10–100 current pulses sustain the arc. When taking photographs with a longer exposure time, instead of a single arc, an overlap of the arc at many subsequent positions can be seen, as shown in Figure 3. The fast, rotational motion of the arc results in a characteristic filament-structured bright primary plasma, observed at the nozzle orifice during the operation of the plasma jet.

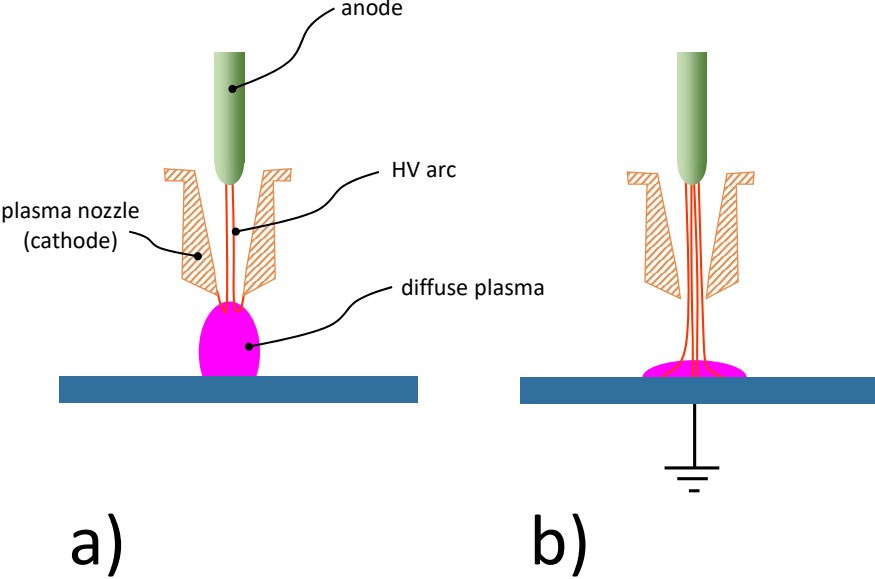

**Figure 4.** The schematically shown plasma jet in contact with (**a**) floating and (**b**) grounded electrically conducting substrate.

The frequency of the HV pulses in the range of tenths of kHz is high enough to avoid complete extinction of the arc between two pulses. The discharge channel remaining along the arc current trace is much easier to reignite than the non-pre-ionized air gap.

The diffuse plasma consists of the products of the interaction between the arc and the gas flowing through it. When the charged particles are confined by electrostatic forces in the core of the arc, neutral particles can freely enter and leave the arc zone, driven by thermodynamic forces. This phenomenon is why the diffuse plasma contains mainly electrically neutral species. However, during the dwell time in the arc, the neutral particles can be energized by electron and photon impacts, resulting in electronically, vibrationally, and rotationally excited species and atomic and molecular radicals, which can transfer the accumulated energy to the substrate surface [30].

If the PAA-PJ is approaching an electrically non-conducting substrate, the diffuse plasma spreads on the substrate surface, but the path of the arcs does not principally change. This is valid also for electrically conducting substrates if they are not grounded. Such a situation is demonstrated in Figure 3b. To reach the transition to the transferred arc operation mode, as displayed in Figure 4b, the substrate must be grounded.

### 3.1.2. Discharge Stability

When operating the plasma generator with CDA, the PAA-PJ is stable for any pulse frequency and gas flow at the power level of 100%. The minimum frequency of 40 kHz selectable on the PS 2000 and the minimum CDA flow of 40 SLM recommended for operation with CDA ensure stable operation. This is not the case for nitrogen–hydrogen gas mixtures. The conditions of stable operation for $N_2$-$H_2$ gas mixtures were investigated, and the results are visualized for 5% and 8% of hydrogen in Figure 5. Each curve represents the border separating the parameter zone of stable operation (right-top from the curve) from the zone of non-stable operation (left-bottom from the curve). With FG 95/5, the minimum setting of the frequency of 40 kHz resulted in stable operation only for a very high gas flow of 70 SLM or more. On the other hand, at a typical gas flow of 50 SLM, the stability can be ensured for pulse frequencies of more than 46 kHz.

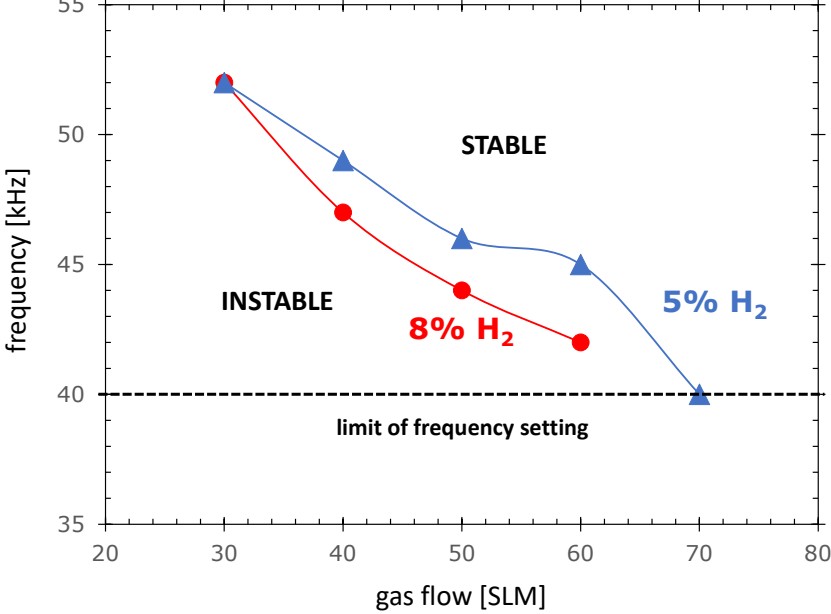

**Figure 5.** The zones of stability and instability of the PAA-PJ with dependence on HV pulse frequency and gas flow for two hydrogen percentages in the $N_2$-$H_2$-gas mixture.

The general tendency is that, with decreasing total gas flow, the discharge becomes less stable, and the frequency must be increased to ensure stability. The possible explanation of this phenomenon can be the increase of the gas temperature in the nozzle with decreasing

gas flow. The consequence of the increasing gas temperature is the increasing diffusivity of the electrons, resulting in faster disruption of the arc channel after the short power pulse. The increase of the frequency allows avoiding the arc disruption, because the next pulse comes faster, before the disruption of the arc occurs.

### 3.1.3. Influence of Hydrogen Percentage on Jet Morphology

For oxide reduction, the hydrogen-containing gas was used. This was the motivation for characterization of the PAA-PJ's operation with gas containing 0 to 10% hydrogen in the nitrogen–hydrogen mixture. Figure 6 shows six pictures of the PAA-PJ with the indicated percentage of hydrogen in the nitrogen–hydrogen gas mixture. From these pictures, it can be concluded that, with the increasing percentage of hydrogen in the gas mixture, both the arc zone and the diffuse plasma zone shrink. The shrinking of the arc zone means the length reduction of the visible part of the arc. The proposed explanation of this phenomenon is based on the increase of the diffusivity and mobility of electrons with increasing percentage of hydrogen, resulting in less drag on the arc by the moving gas and a stronger influence of the electric field on its shape.

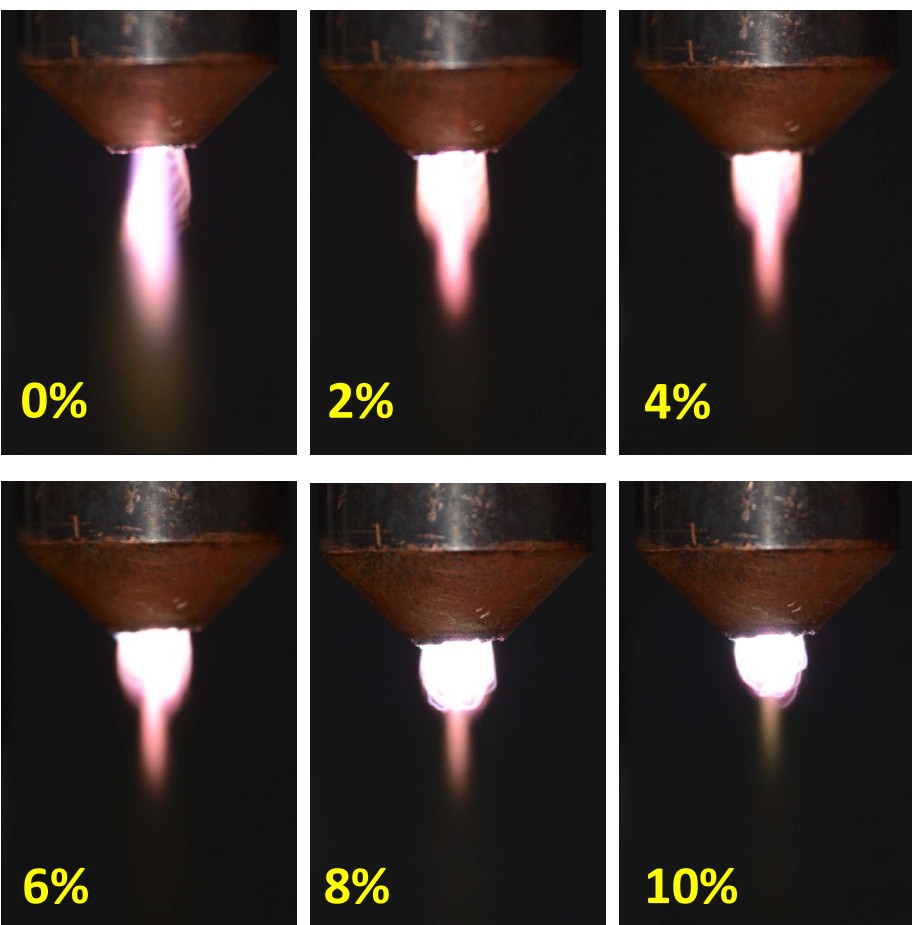

**Figure 6.** The influence of the hydrogen percentage (as depicted in the pictures of [43]) in the nitrogen–hydrogen gas mixture on the plasma jet with no arc transfer in free space. Conditions: total gas flow 60 SLM, frequency 62 kHz, power 100%.

Although the arc length decreases with the hydrogen percentage, the mean voltage drop along the arc increases (see Figure 7) from the 1.2 kV typical for pure nitrogen to 2.5 kV for the gas mixture with 10% of hydrogen. Since the plasma jet power was controlled to be constant by the power source, the mean current flowing through the jet decreased with increasing hydrogen percentage. This means that either the diameter of the arc or the electric conductivity within the arc diminishes. This explanation can give the increase

of the free path for electrons with an increasing concentration of particles with a smaller cross-section for electron impact ionization, resulting in lower ion production rates and, consequently, lower conductivity. The other reason can be the thermal one. Since hydrogen has much higher heat conductivity than nitrogen, the arc should tend to be colder with increasing hydrogen percentage and, consequently, have a higher molecular concentration, resulting in an increased breakdown voltage.

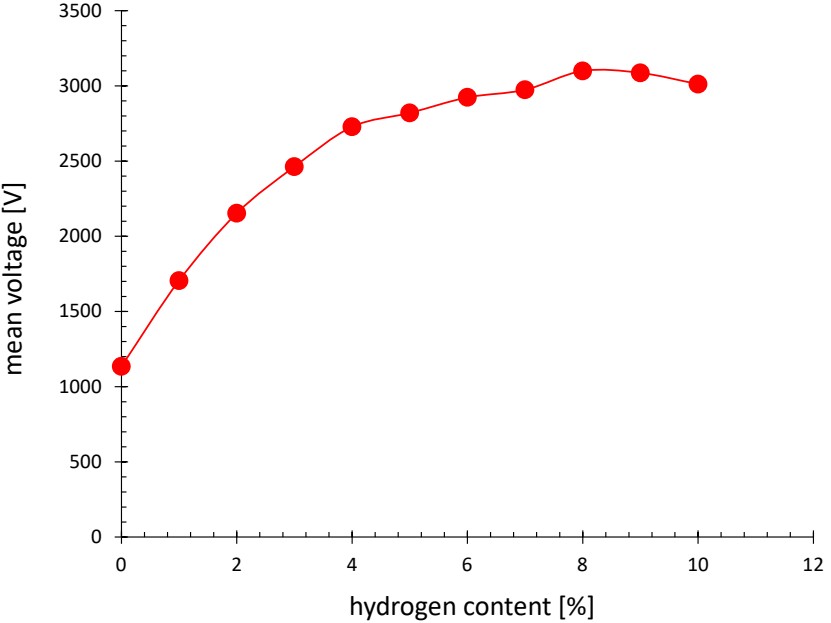

**Figure 7.** The mean voltage of the arc as a function of the percentage of hydrogen in the forming gas for an HV pulse frequency of 65 kHz and a total gas flow of 60 SLM.

### 3.2. Transferred Arc Mode

The HV arcs, which typically end at the nozzle lip, can transfer to the metallic substrate. This can occur only if the substrate is electrically grounded, as shown in Figure 4b. In the diffuse plasma mode, the arc starts at the inner electrode, is expelled several millimeters out of the nozzle, and ends on the nozzle lip. If the nozzle with ignited plasma approaches an electrically conducting and grounded surface or a surface with a large electrical capacity, then the arc transfers to this surface, forming a focused plasma (transferred arc mode plasma). A diffuse plasma, even though present, can no longer be seen, and the arcs are concentrated on a small spot on the substrate.

### 3.2.1. Influence of Hydrogen Percentage on Arc Transition

Since, for the arc transfer to the grounded substrate, the distance from the arc to the substrate should be electrostatically shorter than to the nozzle lip, it can be expected, that the distance for arc transfer will decrease with increasing hydrogen percentage, because the arc zone is shrinking, as discussed in Section 3.1.3. This prediction was confirmed by the investigation of the transition of the jet operation from non-transferred to transferred arc mode [30], summarized in the following.

The transition from diffuse plasma to transferred arc operation mode is not abrupt. When decreasing the distance between the substrate surface and the nozzle, the number of arcs ending on the substrate instead of on the nozzle lip increases gradually. To describe such a transition, the definitions of two distances are required: the distance at which the arc starts to transfer from time to time to the substrate (limit of the purely diffuse mode) and the distance at which the arc ends permanently at the nozzle lip (complete arc transfer). Both transition distances between the nozzle tip and the grounded surface have been determined for nitrogen/hydrogen gas mixtures and are displayed as a function of hydrogen percentage in Figure 8. The upper line in the diagram represents the transition

to the purely diffuse operation mode without arc transfer. The bottom line represents the transition to the purely transferred arc mode. Both lines decrease with increasing hydrogen percentage. This tendency agrees with the shrinking of the arc zone in the PAA-PJ with increasing hydrogen percentage, as demonstrated in Figure 6. The gap between the start of the transfer and complete transfer distances is approximately 3 mm.

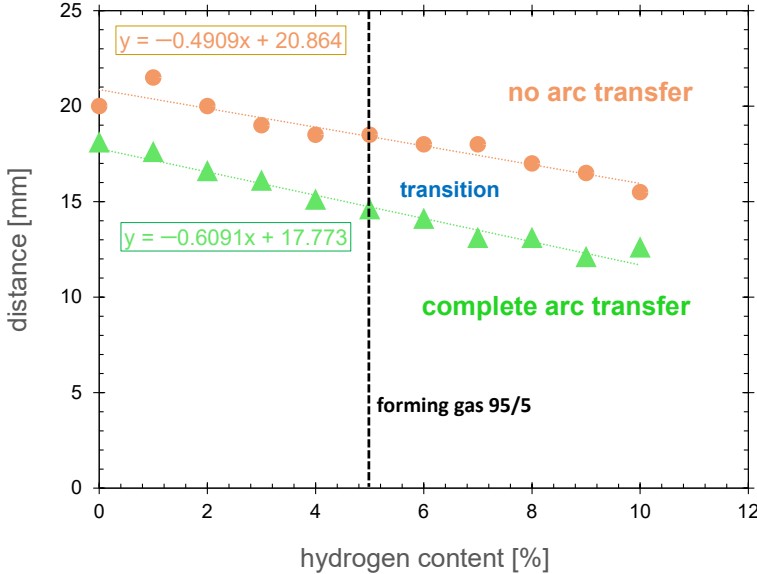

**Figure 8.** The distance between the nozzle tip and the grounded surface at which the transition between the non-transferred and transferred arc operation modes occurs, displayed as a function of hydrogen percentage in the nitrogen/hydrogen gas mixture. These results were determined for an HV pulse frequency of 65 kHz and a total gas flow of 60 SLM.

### 3.2.2. Plasma Focused on Capacitively Grounded Surfaces

If a floating electrode is charged by the current of the transferred arc, the voltage of this electrode increases. After a sufficiently long charging time, the floating electrode potential would reach the mean voltage of the anode of the plasma generator. No voltage difference between the floating substrate and anode would exist anymore, and the current flow would stop. Under such conditions, the arc to the floating electrode is deflected back to the nozzle lip. The charging time of the floating electrode depends on (i) the arc current and (ii) the electric capacity of the floating electrode. If the capacity of the floating electrode is very large, the arc current could flow to it for considerably long time, allowing sustaining the transferred arc. Let us make a simple calculation, showing how this time depends on the geometrical properties of the floating electrode.

Assuming the floating electrode has a surface area $S$ and is separated from a grounded surface by a dielectric with thickness $d$ and dielectric constant $\epsilon_r$, the time $\Delta t$ needed then for charging the electrode up to voltage $\Delta V$ would be expressed by:

$$\Delta t = \frac{\Delta Q}{I} = \frac{C \Delta V}{I} = \epsilon_0 \epsilon_r \frac{S}{d} \frac{\Delta V}{I} \tag{1}$$

Let us consider as a calculation example a conducting surface separated from the ground by a $d = 1$ mm-thick aluminum oxide ($\epsilon_r = 9.5$) plate. Taking the typical mean current of the PAA-PJ of $I = 500$ mA and assuming that the charging up of the floating electrode can increase its potential to no more than $\Delta V = 2820$ V (see the result for 5% hydrogen in Figure 7), Equation (1) delivers charging times depending on the charged area. The results for three areas of the contact pads, as listed in Section 2.1, are summarized in Table 1. These results showed that the charging time even for the largest contact pad was short in comparison with the pulse cycle, which was in the range of 17 μs. Establishing the transferred arc by capacitive grounding of the DC floating electrode would be possible

only with three orders of magnitude larger capacities between the conducting surface and the ground.

**Table 1.** The time needed to charge up the contact pads of different areas to the mean voltage of the anode of 2820 V for a mean arc current of 500 mA.

| Contact Pad Area (mm$^2$) | Time of Charging (ns) |
| --- | --- |
| 8 | 3.8 |
| 32 | 15.2 |
| 175 | 83.0 |

### 3.3. Plasma Bridge

Figure 9 shows the picture of the transferred arc (focused plasma) and the argon plasma bridge focused on the surface of the electrically floating metal plate, taken with a long exposure time. The color difference between the focused plasma and the plasma bridge plasma can be seen. The transferred arc sustained in the forming gas is orange. The plasma bridge, being an argon plasma, is bluish. The distance between the nozzle and the substrate of 6.5 mm was chosen from the distance range for complete arc transfer, as depicted for FG 95/5 in Figure 8. The distance from the injector to the nozzle tip is critical for the transfer of the arc on the substrate. To avoid parasitic arc transfer to the tip of the grounded injector, the radial distance from the nozzle axis was empirically determined to be >3.7 mm. The safe value of 5 mm was chosen for all experiments.

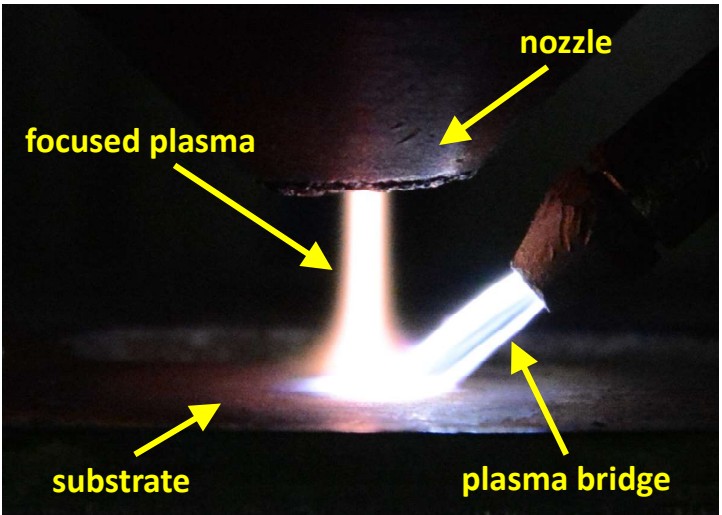

**Figure 9.** Picture of the focused plasma and plasma bridge grounding the substrate. Argon flow: 5 SLM, FG 95/5 flow: 60 SLM, power level: 100%, HV pulse frequency: 62 kHz, distance between the nozzle and the substrate: 6.5 mm.

### 3.3.1. Plasma Bridge Structure

The picture in Figure 9 was taken with a relatively long exposure time, resulting in an overlay of plasma bridge images in different positions. Figure 10 shows two pictures taken with a much shorter exposure time, proving that the plasma bridge changes its start position in the injector orifice. Figure 10a shows the plasma bridge starting at the upper edge of the injector. In Figure 10b, the plasma bridge attached at the bottom part of the injector opening is displayed.

A statistical evaluation based on videos showed that the plasma bridge was much more frequently present in the upper position, even though the lower position is closer to the grounded substrate.

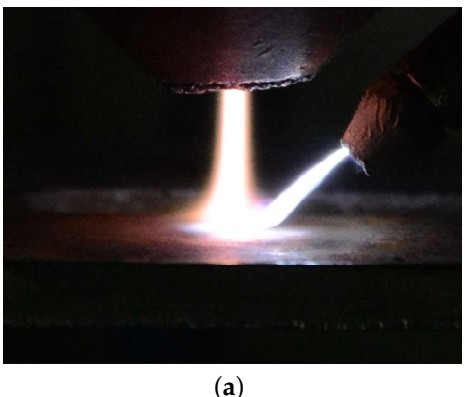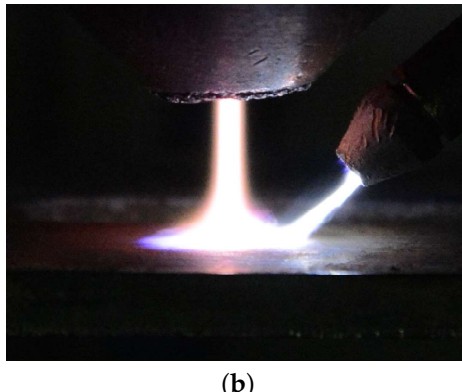

(**a**)                                                                                  (**b**)

**Figure 10.** Pictures of two positions of the plasma bridge start at the injector outlet: (**a**) at the upper side and (**b**) at the bottom side. Other conditions as in Figure 9.

### 3.3.2. Ignition and Extinction by Argon Flow

For better understanding of the plasma bridge, the processes of ignition and extinction were examined in more detail. The first observed regularity was that the plasma bridge did not ignite on the grounded substrate because no voltage across the ignition gap built up. The second regularity was that the plasma bridge could be sustained only to the non-grounded metal electrode on which the arc was transferred. Further, the plasma bridge did not ignite on the free-standing plasma jet. However, the plasma bridge can be sustained on the free-standing plasma jet if it has been previously ignited at the arc-biased substrate. This last property is of high practical importance, as will be shown in Section 3.4.3. The plasma bridge ignition is possible only at some maximum distance between the injector and the substrate. For larger distances, argon becomes too diluted in ambient air, to ensure the easy ignition. This minimum distance increases with increasing argon flow.

The plasma bridge can be initiated and suppressed by switching on and off the argon flow, respectively. The dynamics of these two processes was investigated by taking videos. Figure 11 shows selected frames of these videos, demonstrating changes in the discharge.

Figure 11a shows the situation before the start of argon flow. The arc is not transferred and the plasma bridge not ignited yet. The arc cone is deformed. The asymmetry observed in the arc trajectories can be explained by the electrostatic influence of the grounded injector. The arcs facing the injector are bent more in its direction at a smaller distance from the nozzle than arcs facing away from the injector, because they are attracted by the injector. However, the electric field is not sufficient to cause the gaseous breakdown between the nozzle and the injector in a gaseous environment without argon. The arc transfer and plasma bridge ignition occur in the time between the frame shown in Figure 11a and Figure 11b, one frame later. The time between the frames was 1/60 s. This means that the ignition process's duration was less than 17 ms.

Comparing Figure 11b,c, during the time between these pictures, the intensity of the discharge decreased. The plasma bridge focused on one side of the injector opening. The substrate area covered by plasma diminished. This effect can be explained by the control scheme of the HV generator. It controls voltage and current to keep a constant power. Since the ignition of the plasma bridge results in the abrupt reduction of the load resistivity, and consequently increases the current, the control mechanism reduces the arc voltage to stabilize the power level.

After switching off the argon flow, a lag of the flow exists, but the length of the argon-reach zone shrinks, which can be observed between Figure 11d,e. Furthermore, the strong gas flow from the nozzle drives the residua of the plasma bridge to the right in Figure 11d,e. With advancing dilution of argon in nitrogen, the discharge between the nozzle and the injector converts into a nitrogen arc and, finally, jumps off the injector edge (see Figure 11f).

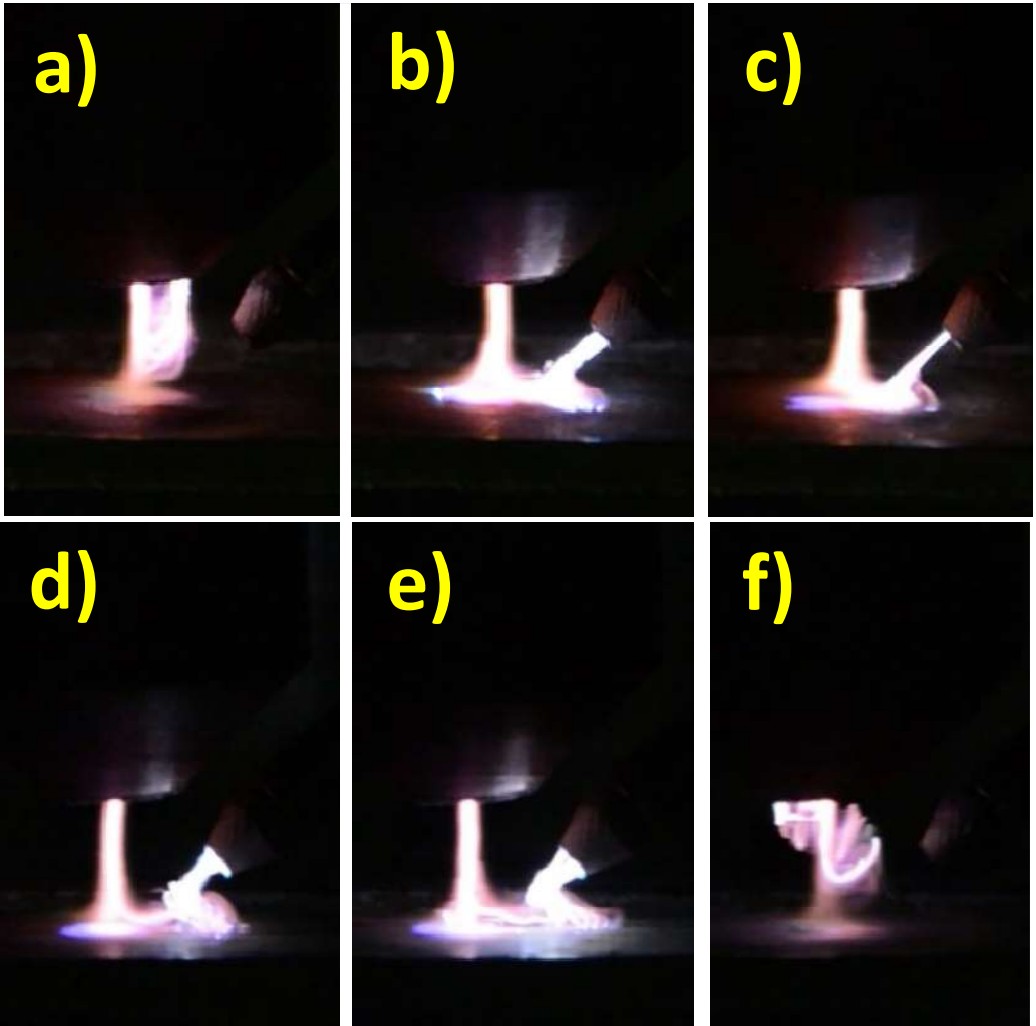

**Figure 11.** Pictures of ignition (**a**–**c**) and extinction (**d**–**f**) of the plasma bridge initiated by the start and stop of the argon flow, respectively. Argon flow: 5 SLM, FG 95/5 flow: 60 SLM, power level: 100%, frequency: 62 kHz, distance between the nozzle and the substrate: 6.5 mm.

### 3.3.3. Plasma Bridge Current

Figure 12 shows the dependence of the mean current flowing from the substrate to the injector through the plasma bridge. The current vanishes for an argon flow lower than 3 SLM and higher than 10 SLM, which correlates visually with the non-existent plasma bridge. At an argon flow below 3 SLM, the argon-enriched zone does not expand sufficiently to short-cut the electric gap between the biased substrate and the injector. The possible reason for the disappearing of the plasma bridge for an argon flow over 10 SLM is the transition of the argon flow from laminar to turbulent, resulting in the disruption of the argon bridge discharge structure. However, other technological factors limit the maximum value of the plasma bridge argon flow far below 10 SLM:

- The consumption of argon is an economic factor. Its flow should be minimized.
- Argon dilutes the PAA-PJ gas, causing a decrease of the hydrogen concentration at the substrate and, consequently, the diminished efficiency of the oxide reduction process. From this point of view, the argon flow should be as low as possible.
- The presence of the argon at the foot of the transferred arc causes an increase of the contact area between the arc and the substrate and, consequently, a lower power density of the arc. The local temperature of the surface decreases, and the process efficiency drops. To minimize this effect, the smallest possible argon flow should be applied.

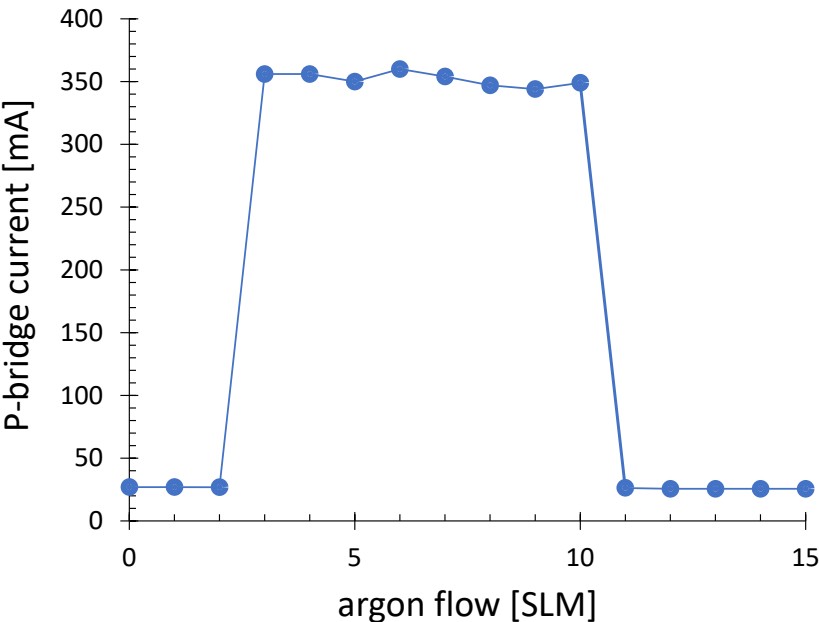

**Figure 12.** The plasma bridge current measured on the wire between the injector and ground as a function of the argon flow. The distance between the nozzle and substrate is 6.5 mm. The distance between the injector and substrate is 2.8 mm. The distance between the injector and the nozzle axis is 5 mm. Operation frequency is 62 kHz. Power level is 100%. FG 95/5 flow is 60 SLM.

Between 3 and 10 SLM, the plasma bridge current remains almost constant. This can be explained by the current-limiting role played by the HV arc. The arc current is not primarily influenced by the argon flow value because the arc has much higher resistance than the plasma bridge discharge. Small changes of the last have only a small impact on the total resistance between the anode and ground.

### 3.4. Application Example

The copper surface of an electronic board contact pad oxidizes during long-term storage. The growth of an oxide film results in the loss of the solderability. To make it solderable again, the oxide must be removed. Typically, this can be performed chemically using fluxes [44]. The drawback of this method is the use of large amounts of solvents and other hazardous chemicals. It is known that a hydrogen-containing atmosphere at an elevated temperature can be used for copper oxide reduction [45] following a simplified chemical formula [46]:

$$Cu_2O + H_2 \longrightarrow 2Cu + H_2O \qquad (2)$$

Essential for the description of the reduction process of oxide layers is the diffusion of hydrogen atoms through the already-reduced copper film. Delivering the required heat and hydrogen atoms by hydrogen plasma is considered a promising approach [47,48]. Several studies demonstrated the reduction of copper oxide thin films by hydrogen-containing atmospheric pressure plasmas. Sawada et al. [49] used an atmospheric pressure glow generated in a dielectric barrier discharge (DBD). Sener et al. [50] showed the application of an atmospheric pressure plasma jet (APPJ) operated with <5 W power with a frequency of 13.56 MHz, and a $He/H_2$ gas mixture. Inui et al. [51] performed this with an $Ar/H_2$ gas mixture. An atmospheric pressure inductively coupled plasma (AP-ICP) microjet was another technique investigated by Tajima et al. [52,53]. Lee et al. [54] used a 100 mm-wide linear beam source operated with a 500 W radio frequency power at 27.12 MHz, 33.0 SLM of argon (Ar), and a 7.0 SLM gas mixture consisting of 5% hydrogen and 95% argon.

Furthermore, the transferred PAA was successfully used to reduce the oxides on electrically grounded silver or copper surfaces [55]. For this purpose, the forming gas

(the mixture of nitrogen and hydrogen) was used as the ionization gas. As explained in Section 3.2, this method failed as for non-grounded substrates. The reduction process conducted with diffuse plasma is on orders of magnitude slower than with transferred arc plasma and, consequently, of no practical interest for this study. To demonstrate the effectiveness of the plasma bridge grounding, the plasma reduction of oxide on the electronic board described in Section 2.1 was used. The task was to make the contact pads on the electronic board free from oxides without damaging contact pads and the solder stopping paint.

### 3.4.1. Oxidation Process

The growth of the native oxide on copper is slow, in the range of a few nanometers per day [56]. The grade of the oxidation on the surface of the contact pads on aged electronic plates varies strongly from plate to plate and from place to place. To ensure the reproducibility and easier evaluation of the reduction process, the electronic plates were treated by the APPJ operated with CDA to grow the oxide film, visible by the color change. The copper oxide formation by thermal oxidation follows the phase sequence [57]:

$$Cu \longrightarrow (Cu + Cu_2O) \longrightarrow Cu_2O \longrightarrow (Cu_2O + CuO) \longrightarrow CuO \qquad (3)$$

First, the brown cuprous oxide ($Cu_2O$) film grows. For temperatures over 330 °C, $Cu_2O$ converts to black cupric oxide (CuO). During Cu oxidation, the Cu cations diffuse outward through the growing $Cu_2O$ layer (the inward diffusion of oxygen ions is much slower) [58]. It is known that the oxide layer on copper is composed of three regions: a thin CuO top layer, a thick $Cu_2O$ layer, and an oxygen-containing transition region. However, the study of the oxidation kinetics of copper at 350 °C to 1050 °C shows that, in this temperature range mainly, $Cu_2O$ grows and the top CuO layer is comparatively thin. The main growth-rate-limiting factor is outward diffusion of Cu through the $Cu_2O$ layer [59].

The oxidation process conditions used in this study are summarized in Table 2. The distance between the nozzle and the contact pad surface was 10 mm. As for reduction, the oxidation process was conducted using plasma bridge grounding. However, the diameter of the transferred arc spot on the metal surface was about 8 mm. The temperature during the oxidation taken from the curve for CDA in Figure 13 was 725 °C. The result of the pre-oxidation is illustrated on the left side of Figure 14.

**Table 2.** The standard process parameters for reduction and pre-oxidation of copper contact pads o an $A_2O_3$ plate.

| Parameter | Reduction | Oxidation |
|---|---|---|
| power level | 100% | 100% |
| nozzle-substrate distance | 12 mm | 10 mm |
| pulse frequency | 60 kHz | 60 kHz |
| speed | 100 mm/s | 100 mm/s |
| plasma gas | FG95/5 | CDA |
| plasma gas flow | 57 SLM | 57 SLM |
| plasma bridge gas | Ar | Ar |
| plasma bridge gas flow | 7 SLM | 7 SLM |
| length of treatment path | 180 mm | 180 mm |
| step between paths | 4 mm | 8 mm |
| number of paths | 24 | 12 |
| number of runs | 3 | 4 |

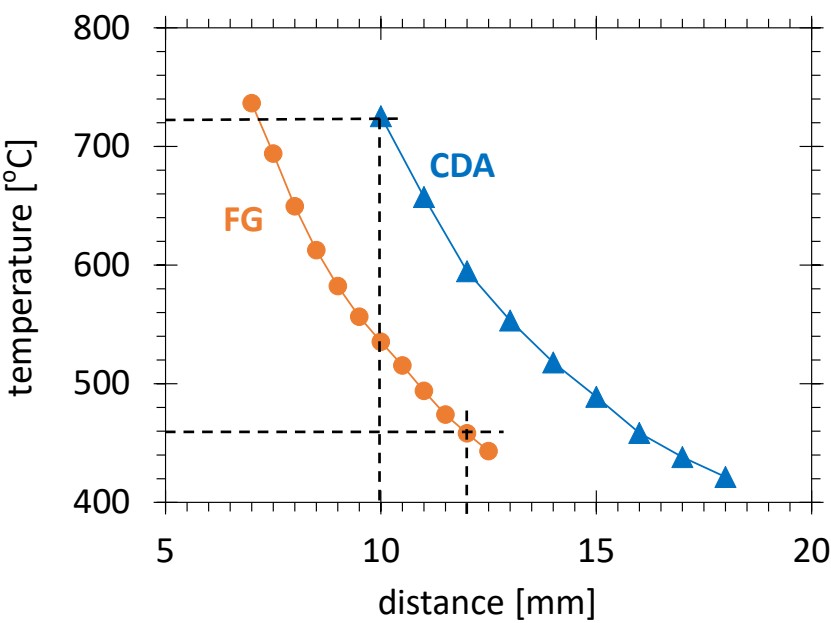

**Figure 13.** The static temperature measured in the FG95/5 and CDA plasma jet, for an HV pulse frequency of 56 kHz and a gas flow of 40 SLM.

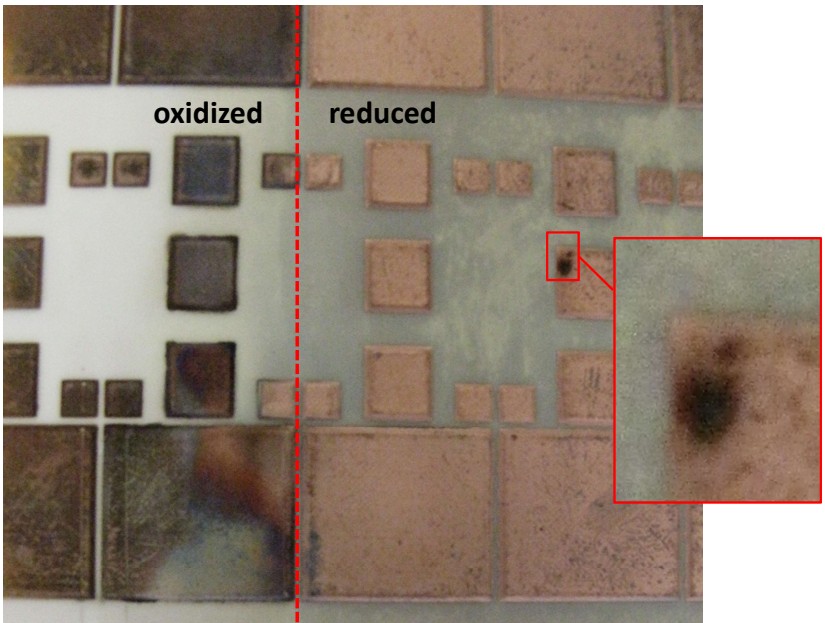

**Figure 14.** The ceramic board with floating copper electrodes pre-oxidized with air plasma (**left side**) and subsequently partially reduced with forming gas 95/5 plasma (**right side**). The electrodes are grounded by the plasma bridge operated with a gas mixture consisting of 98% argon and 2% hydrogen. In the magnification, the oxidized zone due to the accidental extinguishing of the plasma bridge between contact pads.

Using a simplified oxidation model, the thickness of the cuprous oxide film growing during the oxidation process on copper is calculated in Appendix A, and it was about 66 nm. The color of the oxide grown (left side in Figure 14) was darker than for $Cu_2O$, but lighter than for CuO [60].

3.4.2. Reduction Process

Forming gas 95/5 was used for the process development for reasons discussed in Section 2.1. To ensure the most-effective process, the power was set to 100% for all experi-

ments (see Table 2). For maximum process stability, the high HV pulse frequency of 60 kHz and the high total gas flow of 56 SLM were selected. As indicated in Figure 8, to ensure a complete arc transfer to the substrate for FG 95/5, the distance between the substrate and the nozzle should be less than 15 mm. However, to ensure that the plasma bridge does not disappear during the movement between the contact pads, the slightly shorter distance of 12 mm was chosen.

For the given distance between the nozzle and the substrate, the choice of the plasma head speed is a compromise between the chemical process efficiency and the thermal overload of the substrate. The chemical reactivity by reduction grows exponentially with increasing temperature. To increase this temperature for a given distance, power, and frequency, the speed should be minimized.

The typical axial temperature distribution in the diffuse plasma, measured statically for FG95/5 as the ionization gas, is displayed in Figure 13 as a function of the distance between the nozzle tip and the temperature sensor. According to this curve, for the selected distance of 12 mm, the static temperature would reach 460 °C. This temperature does not represent the actual temperature of the alumina plate exposed to the moving plasma plume. The dynamic temperature on the substrate was much lower, but too low a speed would result in an increase of local temperature gradients, inducing thermal–mechanical stress in the electronic board. In extreme cases, the damage of the alumina plate can occur. In less severe cases, the bowing of the electronic board occurs, resulting in parasitic variations of the substrate–nozzle distance and, consequently, nonhomogeneous treatment. Furthermore, hot copper surfaces are prone to back-oxidation if the process is conducted in ambient air. By trial and error, the speed of 100 mm/s was found to be the minimum one at which no such disturbing effects were observed.

The length of the treatment path of 180 mm was longer than the length of the electronic board to ensure that the points of return during the scanning movement of the plasma jet were outside of the electronic board, to avoid the local overheating at these points. The step between lines of treatment was 4 mm, according to the diameter of the transferred arc spot on the metal surface.

### 3.4.3. Treatment Results

The first experiments of the large-area- plasma-bridge-assisted reduction process were conducted with Varigon $H_2$ gas. This gas was chosen for improved reduction of the oxidized surface. However, the breakdown voltage of Varigon is much higher than for argon. As a consequence, from time to time, the extinction of the plasma bridge on the way between two contact pads and a delayed ignition were observed. In such cases, non-reduced zones were left. Moreover, a direct breakdown between the contact pad and the injector through ambient air is possible, leading to the local damage/oxidation of the contact pad surface (see the magnification in Figure 14). The probability of such parasitic bridge discharge decreases with the decreasing breakdown voltage of the plasma bridge gas. To ensure the reliable ignition of the plasma bridge, pure argon was chosen instead of Varigon for the further experiments. The argon flow of 7 SLM (see Table 2) chosen for processing is larger than the value of 5 SLM applied for plasma bridge investigation. The reason is the larger distance between the nozzle and the substrate and, consequently, between the injector and the substrate. To ensure the plasma bridge ignition across a longer spark gap, the zone of high argon concentration must be extended, which was achieved by increasing the flow.

The whole oxidized electronic board was treated with the assistance of the argon plasma bridge. To reach a complete oxide reduction, three runs of the process were needed. This correlates with the simplified model and calculation in Appendix B, showing that the thickness of the reduced film reached after three runs of the process defined in Table 2 was 67 nm, corresponding to the thickness of the entire oxide film.

Despite the non-conducting gaps between the contact pads (see Figure 15), the plasma bridge did not extinguish during the entire duration of the electronic board treatment. This held also for positions outside of the electronic board area.

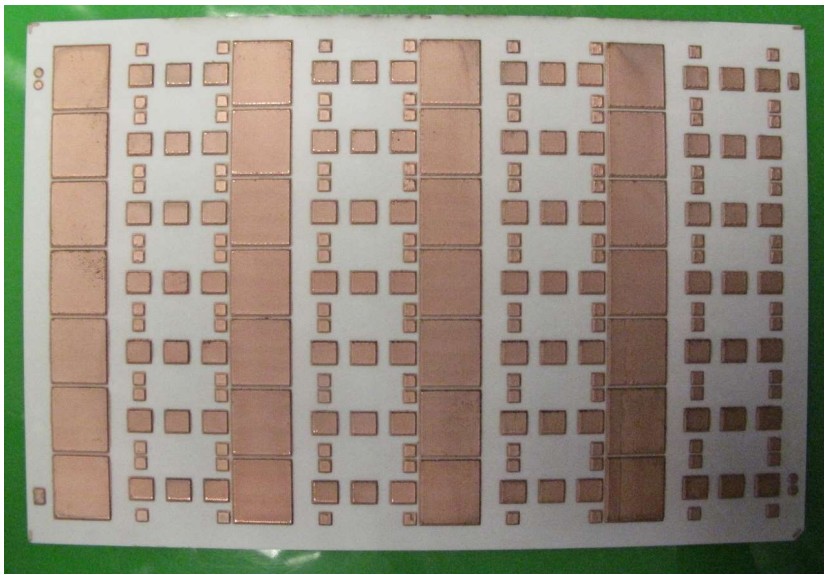

**Figure 15.** An entire $Al_2O_3$ plate with copper electrodes reduced with forming gas 95/5 plasma jet and argon plasma bridge. The treatment conditions are summarized in Table 2.

The success of the reduction was judged on the basis of two effects. The first was the appearance of the contact pad changing from dark brown after pre-oxidation to shiny copper after the reduction process. The second was the change of the wettability. The droplet test with distilled water was applied to determine the contact angle between the droplet surface and the substrate plane. The contact angle measured at the top layer of cupric oxide (CuO) of more than 90° before reduction process agrees with the literature data [61]. It decreased to an unmeasurable small value (complete spreading of the water droplet across the contact pad surface) after the reduction process, which is typical for a clean copper surface.

## 4. Conclusions

If the PAA-PJ is used for the treatment of electrically conducting substrates, the transferred arc can be established only to electrically grounded surfaces. In some cases, the conducting surfaces, e.g., contacting pads of electronic boards, are constructed as electrically floating. In this study, it was shown that, for the grounding of floating electrodes, the plasma bridge ignited in argon can be used. The plasma bridge can be achieved by flooding of the contact pad by argon from an electrically conducting and grounded injector tube. The plasma bridge was ignited because, without grounding, the potential of the electrically floating substrate rose to many hundreds of volts, sufficient for gaseous breakdown in argon between the substrate and the injector. After ignition, the plasma bridge can be sustained even without the substrate between the injector and the HV arc.

The establishment of the plasma bridge was possible for argon flows between 3 and 10 SLM. The plasma bridge cannot be sustained for an argon flow below 3 SLM. At such low flows, the argon flux did not expand sufficiently to reach the substrate and allowed closing the electric gap between the biased substrate and the injector. The plasma bridge cannot be sustained for an argon flow over 10 SLM. The possible reason is the transition of the argon flow from laminar to turbulent, resulting in disruption of the argon bridge discharge.

The reduction of oxidized copper contact pads on alumina plates by forming gas 95/5 plasma was investigated as an application example. To achieve the grounding of the contact pads, the argon plasma bridge was used. The contact pads on the electronic board

were made free from oxides without damage on the electrodes or on the solder stopping paint. Despite the non-conducting gaps between the contact pads, the plasma bridge was not extinguished during the entire duration of the electronic board treatment. The cuprous oxide film with a thickness of about 66 nm on an electric board with an area of 210 cm$^2$ could be reduced within a 2 min process, reaching the processing speed of 4 cm$^2$/s. For the typically much thinner native oxide, the process speed could be increased on order of magnitude.

**Author Contributions:** Conceptualization, S.N., D.K. and M.H.; methodology, S.N., D.K. and M.H.; software, M.H.; validation, D.K. and M.H.; formal analysis, D.K. and M.H.; investigation, D.K. and M.H.; resources, S.N.; data curation, D.K. and M.H.; writing—original draft preparation, D.K. and M.H.; writing—review and editing, D.K.; visualization, M.H. and D.K.; supervision, S.N.; project administration, S.N. All authors have read and agreed to the published version of the manuscript.

**Funding:** This research received no external funding.

**Data Availability Statement:** Data supporting reported results can be obtained on request from the corresponding author.

**Acknowledgments:** The authors thank AB Mikroelektronik GmbH for providing the electronic boards.

**Conflicts of Interest:** The authors declare no conflict of interest.

## Abbreviations

The following abbreviations are used in this manuscript:

| | |
|---|---|
| APP | Atmospheric pressure plasma |
| APPJ | Atmospheric pressure plasma jet |
| PAA | Pulsed atmospheric arc |
| PAA-PJ | Pulsed atmospheric arc plasma jet |
| PET | Polyethylene terephthalate |
| PI | Polyimide |
| PDMS | Polydimethylsiloxane |
| HV | High voltage |
| ICP | Inductively coupled plasma |
| DBD | Dielectric barrier discharge |
| CDA | Compressed dried air |
| FG | Forming gas |
| MFC | Mass flow controller |
| SLM | Standard liters per minute |

## Appendix A. Estimation of the Oxide Layer Thickness

The aim of this appendix is to estimate the thickness of the cuprous oxide film after the oxidation process. For this calculation, the following assumptions were made: (i) the oxidation occurs rapidly at the surface, and the diffusion of copper ions from the bulk to the surface through the oxide (see Figure A1a) is the rate-limiting step [58]; (ii) only cuprous oxide is growing; (iii) the loss of copper ions in the grown oxide layer is neglected; (iv) although the $Cu_2O$/Cu interface shifts from the surface into the inner region, it is regarded as stationary in comparison with the diffusion rate of the copper ions.

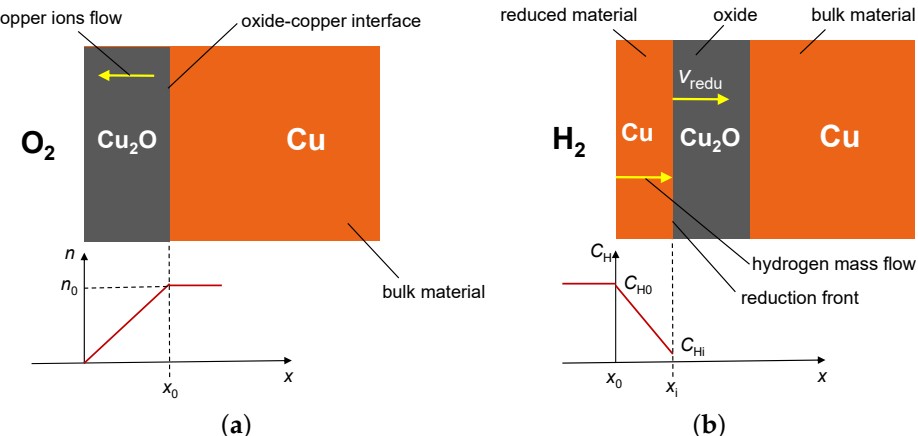

**Figure A1.** Simplified mechanisms of (**a**) oxidation of copper and (**b**) reduction of cuprous oxide.

Under these assumptions, the thickness of the oxide $d_{ox}$ grown after time $t_{ox}$ can be approximated according to the parabolic model of oxidation [60]:

$$d_{ox}(t) = \frac{\sqrt{k_{ox}t_{ox}}}{\rho_{ox}} \tag{A1}$$

where $\rho_{ox}$ is the mass density of the oxide film. Assuming that the oxide consists of cuprous oxide only, the value $\rho_{ox} = 6 \times 10^3 \ \mathrm{kg \cdot m^{-3}}$ was taken. $k_{ox}$ is the parabolic rate constant for oxidation. The dependence of $k_{ox}$ on temperature $T$ follows the Arrhenius equation modified by the influence of the partial pressure of oxygen $p_{O2}$ and temperature $T$ at the substrate surface [58]:

$$k_{ox}(T) = k_{ox0}\left(\frac{p_{O2}T_0}{p_0 T}\right)\exp\left(-\frac{E_{ox}}{RT}\right) \tag{A2}$$

where $k_{ox0}$ is the pre-exponential constant, $E_{ox}$ is the activation energy of the oxidation process, and $R$ is the universal gas constant. $p_0 = 0.1$ MPa and $T_0 = 273.15$ K are the pressure of 1 bar and the absolute temperature of 0 °C, respectively. The pre-exponential constant and the activation energy valid in the temperature range from 350 °C to 1050 °C are $k_{ox} = 1.93 \times 10^{-1} \ \mathrm{kg^2 \cdot m^{-4} \cdot s^{-1}}$. and $E_{ox} = 9.5 \times 10^4 \ \mathrm{J \cdot mol^{-1}}$, respectively [62].

Formula (A1) can be used for the estimation of the oxide thickness after the oxidation process defined by the parameters in Table 2. The static temperature at the oxidized surface taken from the CDA-plot in Figure 13 at a distance of 10 mm between the nozzle and the substrate surface is 725 °C. Due to low heat capacity and high heat conductivity of the thin copper layer of the contact pads, it can be assumed that the oxide film reaches this temperature within the interaction time with the plasma plume. The temperature $T = 725\ °\mathrm{C} + 273.15\ \mathrm{K} = 998.15$ K and the oxygen partial pressure $p_{O2} = 21$ kPa were used for the calculation of the parabolic rate constant:

$$k_{ox}(998.15\ \mathrm{K}) = 1.93 \times 10^{-1} \ \mathrm{kg^2 \cdot m^{-4} \cdot s^{-1}} \times \left(\frac{21\ \mathrm{kPa \cdot 273.15\ K}}{0.1\ \mathrm{MPa \cdot 998.15\ K}}\right)$$

$$\exp\left(-\frac{9.5 \times 10^4\ \mathrm{J \cdot mol^{-1}}}{8.314\ \mathrm{J \cdot K^{-1} \cdot mol^{-1}} \times 998.15\ \mathrm{K}}\right) = \tag{A3}$$

$$4.94 \times 10^{-7} \ \mathrm{kg^2 \cdot m^{-4} \cdot s^{-1}}$$

Taking a plasma jet speed of 100 mm/s and assuming an effective diameter of the plasma plume of 8 mm and four runs of the plasma head, the interaction time of the plasma

plume with the substrate is $t_{ox}$ = 320 ms. Taking this value in Equation (A2), the thickness of the oxide film reached after oxidation process is

$$d_{ox}(320 \text{ ms}) = \frac{\sqrt{4.94 \times 10^{-7} \text{ kg}^2 \cdot \text{m}^{-4} \cdot \text{s}^{-1} \times 0.32 \text{ s}}}{6000 \text{ kg} \cdot \text{m}^{-3}} = 66.3 \text{ nm} \quad \text{(A4)}$$

**Appendix B. Estimation of the Reduced Oxide Thickness**

The aim of this appendix is to estimate the thickness of the reduced oxide film. For this calculation, the following assumptions were made [46]: (i) the reduction at the $Cu/Cu_2O$ interface (see $x = x_i$ in Figure A1b) is rapid, and the diffusion of the hydrogen atoms in the reduced copper layer is the rate-limiting step; (ii) the loss of hydrogen atoms in the reduced copper layer is neglected; (iii) although the $Cu/Cu_2O$ interface shifts from the surface into the inner region, it was regarded as stationary in comparison with the diffusion rate of the hydrogen atoms; (iv) at the surface, the hydrogen concentration in the gas phase was assumed.

Under these assumptions, the thickness of the reduced cuprous oxide film after the treatment time of $t_{redu}$ can be approximated by [46]

$$d_{redu}(t) = \left( \frac{D_H C_{H0}}{C_{Cu2O}} t_{redu} \right)^{1/2} \quad \text{(A5)}$$

where $D_H$ is the diffusivity of hydrogen atoms in copper, $C_{H0}$ is the molar density of hydrogen at the surface, and $C_{Cu2O}$ is the molar concentration of $Cu_2O$ in the copper oxide layer.

$D_H$ is dependent on temperature $T$ according to the Arrhenius equation:

$$D_H(T) = D_{H0} \exp\left( -\frac{E_D}{RT} \right) \quad \text{(A6)}$$

where $D_{H0}$ is the pre-exponential constant, $E_D$ is the activation energy of the diffusion process, and $R$ is the universal gas constant. The pre-exponential constant and the activation energy valid in the temperature range from 720 K to 1200 K are [62] $D_{H0} = 1.131 \times 10^{-6} \text{ m}^2 \cdot \text{s}^{-1}$. and $E_D = 38,850 \text{ J} \cdot \text{mol}^{-1}$, respectively. From the forming gas curve in Figure 13 at a distance of 12 mm from the nozzle, the static temperature of $T = 460 \text{ °C} + 273.15 \text{ K} = 733.15 \text{ K}$ was reached. Due to low heat capacity and high heat conductivity of the thin copper layer, it was assumed that it reaches this temperature during the interaction time with the plasma plume.

$$D_H(733.15 \text{ K}) = 1.131 \times 10^{-6} \text{ m}^2 \cdot \text{s}^{-1} \times \exp\left( -\frac{38,850 \text{ J} \cdot \text{mol}^{-1}}{8.314 \text{ J} \cdot \text{K}^{-1} \cdot \text{mol}^{-1} \times 733.15 \text{ K}} \right) =$$
$$1.93 \times 10^{-9} \text{ m}^2 \cdot \text{s}^{-1} \quad \text{(A7)}$$

The molar density of $Cu_2O$ is

$$C_{Cu2O} = \rho_{ox}/M_{Cu2O} \quad \text{(A8)}$$

Using for cuprous oxide the mass density $\rho_{ox} = 6 \times 10^3 \text{ kg} \cdot \text{m}^{-3}$ and the molar weight $M_{Cu2O} = 143.1 \times 10^{-3} \text{ kg} \cdot \text{mol}^{-1}$, the molar density of $C_{Cu2O} = 4.19 \times 10^4 \text{ mol} \cdot \text{m}^{-3}$ is obtained.

The molar density of hydrogen atoms at the surface is given by the formula:

$$C_{H0} = \frac{\eta T_0}{V_{m0} T} \quad \text{(A9)}$$

where $\eta = 5\%$ is the molar percentage of hydrogen in the forming gas FG95/5 and $V_{m0} = 22.71 \times 10^{-3} \, \text{m}^3 \cdot \text{mol}^{-1}$ is the molar volume of hydrogen at $T_0 = 273.15$ K. With these values, Equation (A9) gives a molar density $C_{H0} = 0.82 \, \text{mol} \cdot \text{m}^{-3}$.

Taking the plasma jet speed of 100 mm/s and assuming the effective diameter of the plasma plume of 4 mm and three runs of the plasma head, the interaction time of the plasma plume with the substrate is $t_{\text{redu}} = 120$ ms.

Applying this $t_{\text{redu}}$ together with the calculated diffusivity and molar densities in Equation (A5) gives the thickness of the reduced layer of

$$d_{\text{redu}}(120 \text{ ms}) = \left( \frac{1.93 \times 10^{-9} \, \text{m}^2 \cdot \text{s}^{-1} \times 0.82 \, \text{mol} \cdot \text{m}^{-3}}{4.19 \times 10^4 \, \text{mol} \cdot \text{m}^{-3}} 120 \text{ ms} \right)^{1/2} = 67.3 \text{ nm} \quad \text{(A10)}$$

Equation (A5) is valid only if the molar density of hydrogen is lower than the solubility of hydrogen in copper. Otherwise, the diffusion would not be the process rate-limiting step. The molar solubility of hydrogen in copper is given as

$$S(T) = S_0 \exp\left( -\frac{E_S}{RT} \right) \quad \text{(A11)}$$

Using the activation energy $E_S = 11,130 \, \text{J} \cdot \text{mol}^{-1}$ and the pre-exponential constant $S_0 = 224 \, \text{mol} \cdot \text{m}^{-3}$ [63] the molar solubility is:

$$S_s(460 \, ^\circ\text{C}) = 224 \, \text{mol} \cdot \text{m}^{-3} \times \exp\left( -\frac{11,130 \, \text{J} \cdot \text{mol}^{-1}}{8.314 \, \text{J} \cdot \text{mol}^{-1} \cdot \text{K}^{-1} \times (273.15 + 460) \, \text{K}} \right) = \quad \text{(A12)}$$
$$36.1 \, \text{mol} \cdot \text{m}^{-3}$$

This value is two orders of magnitude higher than the concentration in the gas phase. Consequently, it is consistent with the applied diffusion model.

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
