# Peer review of "Application of Plasma Bridge for Grounding of Conductive Substrates Treated by Transferred Pulsed Atmospheric Arc"

_plasma, doi:10.3390/plasma6010012_

Round 1
Reviewer 1 Report
First of all, I'm very sorry. My research is not in this field, but I think it is a good paper after reading the article and content. There should be many people interested in it.
The article lists the experimental methods and relevant theories, which are helpful to those who are engaged in this field. The whole article format is reasonable, and the English expression is also very clear. I haven't found any problems for the time being. The editor is requested to make a decision based on the opinions of other reviewers.
Author Response
Thank you for encouraging words.
Reviewer 2 Report
Letter to the editor in chief
Dear prof. Dr. Takayoshi Kobayashi
I have read manuscript number: plasma-2193414 with the title:" Application of plasma bridge for grounding of conductive substrates treated by transferred pulsed atmospheric arc", and I found that the authors are The establishing of the plasma bridge is possible for argon flows between 3 and 10 SLM. The reduction of oxidized copper contact pads on alumina plates by forming gas 95/5 plasma was investigated as an application example. I would like the authors to consider the following points before I recommend the publication
1-The abstract of the manuscript must be rewritten in a compact form. The authors must mention the main findings.
2-The introduction must be rewritten.
3-Authors wrote content for the article, why?
4-All symbols in equations must be defined.
5-The presentation and grammar need improvement.
Author Response
Response:
1 – The abstract is rewritten in a compact form. The main findings are mentioned. The important values are added.
2 – The introduction is rewritten to make it shorter and clearer.
Some relevant citations are added.
3 – The content list is removed.
4 – The errors in the description of the equation are corrected. The calculations of the oxide film thickness and the reduction film thickness are added as Appendix.
5 – The presentation and grammar are improved. In detail, the Figures 2, 10 and 13 are improved, the subject of film oxidation is added, and new references are added especially covering the subject of pre-oxidation and reduction of copper surface.
Reviewer 3 Report
Dear Authors
The manuscript is focused on the physical, geometrical, and electrical conditions of the arc transfer to electrically floating surfaces.
The manuscript presented concerns an interesting and actual subject.
The following suggestion and comments should be taken:
1. The overall English needs to be improved. Please seek guidance from a native English speaker if possible ("the" "a", commas, plural form and others could be corrected).
2. The authors could insert more numerical data into the Abstract for enhancement of the manuscript.
3. Due to the number of acronyms used in this manuscript, authors must include a list of abbreviations before the introduction.
4. Figure 2 please correct this image (description) for better quality.
5. Could the authors include the standard deviation of the analysis?
6. Why author choose these systems for the study? Please explain.
Author Response
Response:
- The manuscript is corrected by MDPI English service.
- The abstract is rewritten, and more numerical data is added.
- The list of abbreviations and acronyms is shifted before the introduction.
- In Figure 2 the descriptions are added for better quality.
- The text “This system is especially suitable for this investigation and process, thanks to the positive polarization of the inner electrode, allowing for the cathodic cleaning~\cite{Sarrafi-2010} of the grounded substrates.” is added for justification of the system use.
Reviewer 4 Report
The surface treatment of materials are important for industrial use. In this manuscript, Korzec et al demonstrated that plasma is useful for manufacturing metal. This manuscript is well-organized; however, following points should be clarified.
Major point
#1: How did authors analyze the surface? In figure 14, photograph is shown; however, it is hard to understand what did plasma change?
Minor points
##1: It is too descriptive and hard to understand the advantage of irradiation of plasma.
Author Response
Response to criticism:
#1: The figure caption is extended to make clear what the picture shows. The text referring to the Figure 14 is rewritten.
##1: A sections addressing the mechanism of the Cu2O film growths and reduction are added. The appendix is added with simple model calculations of the oxide film thickness and reduction speed.
Round 2
Reviewer 3 Report
I have not any comments. Now the manuscript is ok.
Reviewer 4 Report
Authors revised manuscript enough.